# The Protective Role of Mitochondria-Associated Endoplasmic Reticulum Membrane (MAM) Protein Sigma-1 Receptor in Regulating Endothelial Inflammation and Permeability Associated with Acute Lung Injury

**DOI:** 10.3390/cells13010005

**Published:** 2023-12-19

**Authors:** Zahra Mahamed, Mohammad Shadab, Rauf Ahmad Najar, Michelle Warren Millar, Jashandeep Bal, Traci Pressley, Fabeha Fazal

**Affiliations:** Department of Pediatrics (Neonatology), Lung Biology and Disease Program, University of Rochester School of Medicine and Dentistry, Rochester, NY 14642, USA; zahra_mahamed@urmc.rochester.edu (Z.M.); mohammad_shadab@urmc.rochester.edu (M.S.); rauf_najar@urmc.rochester.edu (R.A.N.); michelle_millar@urmc.rochester.edu (M.W.M.); jashandeep_bal@urmc.rochester.edu (J.B.); traci_pressley@urmc.rochester.edu (T.P.)

**Keywords:** sigma 1 receptor, endothelial cells, mitochondria-associated endoplasmic reticulum membranes, inflammation, permeability, acute lung injury, immunoglobulin-binding protein

## Abstract

Earlier studies from our lab identified endoplasmic reticulum (ER) chaperone BiP/GRP78, an important component of MAM, to be a novel determinant of endothelial cell (EC) dysfunction associated with acute lung injury (ALI). Sigma1R (Sig1R) is another unique ER receptor chaperone that has been identified to associate with BiP/GRP78 at the MAM and is known to be a pluripotent modulator of cellular homeostasis. However, it is unclear if Sig1R also plays a role in regulating the EC inflammation and permeability associated with ALI. Our data using human pulmonary artery endothelial cells (HPAECs) showed that siRNA-mediated knockdown of Sig1R potentiated LPS-induced the expression of proinflammatory molecules ICAM-1, VCAM-1 and IL-8. Consistent with this, Sig1R agonist, PRE-084, known to activate Sig1R by inducing its dissociation from BiP/GRP78, blunted the above response. Notably, PRE-084 failed to blunt LPS-induced inflammatory responses in Sig1R-depleted cells, confirming that the effect of PRE-084 is driven by Sig1R. Furthermore, Sig1R antagonist, NE-100, known to inactivate Sig1R by blocking its dissociation from BiP/GRP78, failed to block LPS-induced inflammatory responses, establishing that dissociation from BiP/GRP78 is required for Sig1R to exert its anti-inflammatory action. Unlike Sig1R, the siRNA-mediated knockdown or Subtilase AB-mediated inactivation of BiP/GRP78 protected against LPS-induced EC inflammation. Interestingly, the protective effect of BiP/GRP78 knockdown or inactivation was abolished in cells that were depleted of Sig1R, confirming that BiP/GRP78 knockdown/inactivation-mediated suppression of EC inflammation is mediated via Sig1R. In view of these findings, we determined the in vivo relevance of Sig1R in a mouse model of sepsis-induced ALI. The intraperitoneal injection of PRE-084 mitigated sepsis-induced ALI, as evidenced by a decrease in ICAM-1, IL-6 levels, lung PMN infiltration, and lung vascular leakage. Together, these data evidence a protective role of Sig1R against endothelial dysfunction associated with ALI and identify it as a viable target in terms of controlling ALI in sepsis.

## 1. Introduction

Acute lung injury and its most severe form, acute respiratory distress syndrome (ALI/ARDS), are a common cause of respiratory failure in critically ill patients and are associated with an unacceptably high mortality rate of 30–40% in reported cases. Most recently, ALI/ARDS was the major cause of death (~70% mortality rate) in COVID-19 patients (CDC). So far, all current therapies for ALI/ARDS rely on supportive care, and no effective therapeutic options are available to improve the clinical outcome. ALI can occur as a result of infection, trauma, and aspiration; however, the highest number of deaths is observed in patients with sepsis and pneumonia [1,2,3,4]. All these etiologies are, for the most part, associated with vascular inflammation and a loss of integrity in the vascular endothelium with a subsequent accumulation of fluid in alveolar airspace [5,6,7]. Thus, vascular endothelial dysfunction is of central importance in this disease process and requires therapies that are directed toward curbing inflammatory pathways and restoring the integrity of the endothelium. 

Emerging evidence indicates that organelles within the cell engage in extensive communication either directly or indirectly through membrane contacts. Inter-organelle communication is vital for cell function and tissue and organismal homeostasis [8,9,10,11]. Coordination between the endoplasmic reticulum (ER) and the mitochondrion (MITO) plays a pivotal role in numerous cellular processes, such as maintaining Ca^2+^ homeostasis, assembling inflammasomes, regulating mitochondrial dynamics, managing ER stress, promoting cell survival, and influencing lipid metabolism. This intricate communication occurs through a specialized zone of close proximity (approximately 10–25 nm) between the ER and the mitochondria, known as the mitochondria-associated ER membrane [9,11,12,13]. Approximately 5–20% of the OMM (outer mitochondrial membrane) is estimated to be associated with ER membranes, and more than 1000 proteins reside in the MAM region. Disturbance in MAM integrity is correlated with the pathogenesis of metabolic diseases, such as cancer, obesity, diabetes and neurodegenerative disorders, such as Alzheimer’s disease (AD), Parkinson’s disease (PD) and Amyotrophic Lateral Sclerosis (ALS) [14,15,16,17,18]. 

Sigma receptors are bonafide MAM proteins and are predominantly expressed in the central nervous system (CNS), as well as in tissues, such as the heart, liver and lung. Two Sigma receptor subtypes, Sig 1R and Sig 2R, have been identified; however, only Sig1R has been cloned, and the 223 amino-acid-long proteins do not share amino acid homology with any other mammalian proteins. Sig1R has been identified as a non-opioid, Ca^2+^-sensitive, ligand-operated, ER resident chaperone. Sig1R can bind to and modulate a large number of client proteins, including ion channels in both ER and plasma membrane and is, therefore, considered a multidimensional modulator of cellular homeostasis [19,20,21,22,23]. Sig1R dysfunction is implicated in several neurologic and metabolic diseases [24,25,26,27,28]. In the CNS, Sig1R has been implicated in the regulation of the immune activity of microglia and cell fate. The targeting of Sig1R has been shown to impact cytokine production when using immune cells in vitro. Sig1R ligands include psychotropic and neuroprotective agents, and many of them are currently in clinical use, placing Sig1R as an attractive therapeutic target [29,30,31,32,33].

Studies have shown that the functionality of Sig1R is dependent on its interaction with ER chaperone and MAM protein BiP/GRP78. Sig1R is associated with BiP/GRP78, and this binding occurs between the C-terminal domain of Sig1R and the nucleotide-binding domain of BiP/GRP78 [34]. Sig1R agonist (+)-pentazocine causes Sig1R and BiP/GRP78 to dissociate, whereas antagonist haloperidol stabilizes the Sig1R/BiP complex [35,36,37]. Previous studies from our lab using proinflammatory agonist thrombin have identified BiP/GRP78 as a novel regulator of EC dysfunction associated with ALI. Here, we provide evidence that Sig1R plays a protective role in both primary EC and in a mouse model of sepsis against ALI. In addition, the protective effect of Sig1R is mediated via its dissociation from BiP/GRP78. 

## 2. Materials and Methods

**Reagents**: Lipopolysaccharide from Escherichia coli O111:B4 (L2880) was acquired from Sigma-Aldrich (St. Louis, MO, USA). We acquired human alpha thrombin (cat. HT 1002a) from Enzyme Research Laboratories (South Bend, IN, USA). Antibodies for VCAM-1 (sc-13160), ICAM-1 (sc-8439), IkBα (sc-371), RelA/p65 (sc-8008), and β-actin (sc-47778) were acquired from Santa Cruz Biotechnology (Dallas, TX, USA). Antibodies for Sigma1R (61994) and BiP/GRP78 (3177) were procured from Cell Signaling Technology (Beverly, MA, USA). PRE-084 and NE-100 were purchased from Tocris Bioscience (Minneapolis, MN, USA). Fluo-4AM was purchased from Invitrogen (Waltham, MA, USA). Subtilase AB was purified as previously described [38]. An In Vitro Permeability Assay kit (ECM 644) was obtained from Millipore Sigma (Burlington, MA, USA). 

**Endothelial cells**: Human pulmonary artery endothelial cells (HPAECs) were acquired from Lonza (Walkersville, MD, USA). HPAECs were cultured with endothelial basal medium 2 (EBM2) supplemented with bullet kit additives (including 10% FBS) in 2% gelatin-coated flasks. All experiments were performed using cells at passage 6 or below. 

**RNAi knockdown:** Non-targeting control siRNA (siControl), Sigma1R siRNA (siSig1R), and BiP/GRP78 siRNA (siBiP) were purchased from Horizon Discovery (Lafayette, CO, USA). HPAECs were transfected with siSig1R, siBiP, or siControl using DharmaFect1 siRNA transfection reagent from Horizon Discovery (Lafayette, CO, USA), as described previously. Briefly, a mixture of 50 –100 nM siRNA and DharmaFect1 was prepared and added to cells at 60–70% confluency. Experiments were conducted using cells forty-eight hours post transfection.

**Immunoblot analysis:** Following treatments [39], HPAECs were lysed in radio-immuno precipitation assay (RIPA) lysis buffer [1% Triton-X 100,150 mM NaCl, 0.25 mM EDTA (pH 8.0), 50 mM Tris-HCl (pH 7.4), 1% deoxycholic acid, 1 mM sodium orthovanadate, and 5 mM NaF with protease inhibitor cocktail from Sigma-Aldrich (St. Louis, MO, USA). Total cell lysates were separated in SDS-PAGE gel and then transferred onto nitrocellulose membranes. Next, membranes were incubated with primary antibodies overnight at 4 °C followed by incubation with secondary antibodies at room temperature for 1 h. Blots were then developed using an enhanced chemiluminescence (ECL) method, as previously described. It should be noted that the blots presented in the results section may have been taken from membranes containing multiple samples from different experimental groups.

**ELISA:** Cytokine IL-6 and chemoattractant IL-8 levels in HPAEC culture supernatants were determined using Duoset ELISA kits (R&D Systems; Minneapolis, MN, USA) according to the manufacturer’s recommendations [40]. 

**Assessment of RelA/p65 nuclear translocation and DNA Binding:** After treatment, the cells were washed twice with ice-cold phosphate-buffered saline (PBS) followed by resuspension in 400 µL of Buffer A (10 mM KCl, 0.1 mM EDTA, 10 mM HEPES (pH 7.9), 0.1 mM EGTA, 0.5 mM PMSF, and 1 mM DTT). After a 15-min incubation period, NP-40 was added to samples at a final concentration of 0.6% and then centrifuged to collect the supernatant containing cytoplasmic proteins. The nuclei pellet was resuspended in 50 µL of Buffer B (0.4 M NaCl, 1 mM EDTA, 1 mM EGTA, 20 mM HEPES (pH 7.9), 1 mM PMSF, and 1 mM DTT) and incubated on ice for 30 min. Afterward, the lysates were centrifuged to collect supernatants with nuclear proteins [41]. To assess the DNA-binding activity of RelA/p65, we used a DNA binding assay kit from Cayman Chemical (Ann Arbor, MI, USA) and followed the manufacturer’s recommendations [41]. 

**Measurement of endothelial permeability:** Using an electrical cell–substrate impedance sensing (ECIS) system purchased from Applied Biophysics (Troy, NY, USA), trans-endothelial resistance (TER) was monitored to assess endothelial barrier integrity across a confluent monolayer [42]. Briefly, HPAECs were grown on 2% gelatin-coated gold microelectrodes to confluence in EBM-2 media supplemented with bullet kit additives and 10% FBS. After 24 h, the culture medium was substituted with EBM-2 containing 1% FBS. Following a two-hour incubation period, the cells were subjected to various treatments as dictated by the experimental design. The TER was monitored over several hours and normalized to baseline resistance value.

**In vitro permeability assay:** HPAECs either transfected siControl or siSig1R for 48 h or pretreated for one hour with 10 µM PRE-084 were plated onto trans-well inserts at a density of 20,000 cells/insert followed by cultured to confluence for 48 h. Subsequently, the confluent cell monolayer was challenged with thrombin (5 U/mL) for thirty minutes. After, a FITC-Dextran permeability test was performed to determine monolayer integrity. The permeation process was halted by extracting the trans-well inserts from the wells. The media in the receiver tray was transferred to a 96-well opaque plate to measure fluorescence. Fluorescence intensity was quantified using a SpectraMax M5 plate reader (Molecular device, San Jose, CA, USA) with 485 nm excitation and 535 nm emission filters. 

**Intracellular Ca^2+^ measurements using Fluo-4 AM:** Human pulmonary artery endothelial cells (HPAECs) were grown to confluence on a gelatin-coated 35 mm glass coverslip bottom dish (MatTek Life Sciences). The cells were loaded with 5 μM Fluo-4 AM (Invitrogen) in serum-free MCDB 131 (Corning) at 37 °C for 25 min. After dye loading, the cells were washed three times and placed on stage in calcium-free imaging buffer (136 mM NaCl, 4.7 mM KCl, 10 mM HEPES, 1 mM EGTA, 5.5 mM glucose, 1 mM Na_2_HPO_4_, and 0.56 mM MgCl_2_-6H_2_0; pH 7.4). Images were captured every 3 s at excitation of 488 nM using an inverted Axio Imager M2m confocal microscope (Zeiss, Jena, Germany) equipped with 40× oil objective. Following 30-s baseline recording, cells were stimulated with thrombin (5U/mL). Fluo-4 AM fluorescence intensity was quantified using Image J, software, version 1.54. Tracings are representative of mean Fluo-4 AM fluorescence intensity.

**Mouse model of ALI:** Cecal Ligation and Puncture (CLP) Male 8 to 10 wk-old C57BL/6 mice (wild-type, WT; Jackson laboratory) were fasted 16 hr prior to CLP. The mice were sedated using 2.5% isoflurane. Under sterile conditions, a small incision (1–2 cm) was made to the lower left abdominal cavity. Using forceps, the cecum was gently pulled out, ligated with sterile 3-0 silk suture approximately 0.9 cm proximal to the cecum, and punctured through-and-through up to eight times with an 18-gauge needle. A small amount of feces was squeezed out of the holes into the peritoneal cavity to start a strong microbial infection. Cecum was gently pushed back into the cavity without spreading feces on the abdominal wall, and the incision site was closed with 5.0 ethilon silk sutures. The mice were resuscitated with 3 mL/g body weight normal saline and returned to the cage with unrestricted access to both wet and dry food as well as water. Twelve hours after CLP, the mice underwent anesthesia with pentobarbital sodium (50 mg/kg) for additional testing, followed by euthanasia. The University of Rochester Committee on Animal Resources (UCAR) approved all animal care and handling. All experiments strictly adhered to the guidelines provided by the National Institutes of Health.

**Evaluating pulmonary inflammation and injury:** An hour prior to euthanasia, mice received a retro-orbital injection of Evan’s blue dye (EBD; 30 mg/kg) [43]. Subsequently, 10 mL of PBS with 5 mM EDTA was perfused gently through the right ventricle to clear the blood. Next, all four lobes of the right lungs were harvested, blotted dry, and promptly snap-frozen in liquid nitrogen. To calculate the wet-to-dry lung weight ratio, the left lungs were weighed before and after being dried at 60 °C for 24 h. The dried lung samples were subjected to a 24 h incubation at 60 °C in formamide, followed by centrifugation. The absorbance of the supernatant was measured at 620 nm and 740 nm. To account for any tissue heme pigment contamination, absorbance readings at 740 nm were used for correction. The concentration of EBD in the lung tissue was determined via comparison to a standard curve and expressed in micrograms of EBD per gram. Mouse lung homogenates from the superior lobe were lysed in RIPA buffer supplemented with a protease inhibitor cocktail. The levels of ICAM-1 and IL-6 were determined in lung homogenates via ELISA. Polymorphonuclear neutrophil (PMN) recruitment was determined by monitoring the myeloperoxidase activity in lung homogenate from the inferior lobe, as described previously [44]. 

**Statistical analysis:** All data were analyzed using GraphPad Prism 9.0 (San Diego, CA, USA). Two-group comparisons were evaluated using an unpaired, two-tailed Student’s *t* test. For multiple group comparisons, data were analyzed using one-way ANOVA. A post hoc Tukey’s test was used to determine the significance between the groups. Data are presented as mean ± SE. Statistical significance between two groups is denoted by asterisks (* *p* ≤ 0.05, ** *p* < 0.01, *** *p* < 0.001, **** *p* < 0.0001). [42].

## 3. Results

### 3.1. Sigma1R Depletion Potentiates LPS-Induced Inflammatory Responses in ECs

To investigate the involvement of Sig1R in EC inflammation, we employed an RNA interference (RNAi) strategy to diminish Sig1R levels in ECs. The cells underwent transfection with Sig1R-specific siRNA or control siRNA, and Sig1R levels were assessed through the use of immunoblotting. The results revealed a significant reduction in Sig1R levels in cells transfected with Sig1R siRNA compared to those transfected with control siRNA (Figure 1A,B). The same experimental conditions were maintained in subsequent trials involving siRNA-mediated Sig1R depletion. The approach effectively potentiated LPS-induced expression of vascular cell adhesion molecule-1(VCAM-1) and intercellular adhesion molecule-1 (ICAM-1) expression (Figure 1A,C,D). We also determined the effect of Sig1R knockdown on chemoattractant cytokine interleukin-8 (IL-8), another LPS-responsive proinflammatory gene in ECs. A marked increase in LPS-induced IL-8 levels was observed in cells transfected with Sig1R siRNA (Figure 1E). These data highlight the important role of Sig1R in protecting endotoxin-induced inflammation in ECs. 

### 3.2. Sigma1R Activation Protects against LPS-Induced Inflammatory Responses in ECs 

To further establish the protective role of Sig1R in EC inflammation, we took an opposing approach and used Sig1R agonist PRE-084 [45,46], which is known to activate Sig1R by causing its dissociation from ER chaperone BiP/GRP78. BiP/GRP78 coupling to Sig1R is reduced approximately 10 times in the presence of 0.3 µM PRE-084 in CHO cells [22]. The pretreatment of ECs with PRE-084 showed significant inhibition of LPS-induced expression of proinflammatory mediators (ICAM-1, VCAM-1, IL-8 and IL-6), confirming the protective role of Sig1R in mediating EC inflammation (Figure 2A–E). We also monitored the effect of PRE-084 on thrombin-induced proinflammatory gene expression. Thrombin-induced VCAM-1 and IL-8 expression was also significantly reduced in ECs pretreated with PRE-084 (Figure 2F–H). In order to validate the specificity of PRE-084 towards Sig1R, we compared the effect of PRE-084 in control versus Sig1R-depleted cells. PRE-084 failed to block LPS-induced proinflammatory gene expression in Sig1R-depleted cells as compared to the control cells (Figure 3A,B), confirming the specificity of PRE-084 for Sig1R. In line with its role as a Sig1R antagonist, NE-100 [24,45,47], which inactivates Sig1R by blocking its dissociation with BiP/GRP78, failed to have any effect on LPS-induced VCAM-1 and IL-6 expression (Figure 4A–C). These results show that Sig1R’s protective effect on EC inflammation is mediated via its dissociation from BiP/GRP78. 

### 3.3. Sigma1R Depletion Potentiates LPS-Induced Inflammatory Responses in ECs by Inducing IκBα Degradation and Subsequent RelA/p65 (NF-κB) Nuclear Translocation 

Since NF-κB is an essential transcriptional regulator of inflammatory genes encoding ICAM-1, VCAM-1 and IL-8, we next evaluated whether the induced expression of these genes upon Sig1R knockdown is due to the increased translocation of RelA/p65 to the nucleus. Analysis of the nuclear extracts via immunoblotting showed that the depletion of Sig1R potentiated RelA/p65 nuclear uptake and subsequent binding to DNA by LPS (Figure 5A–C). Next, we examined the event upstream of NF-κB nuclear translocation, which involves IκBα degradation in the cytosol. Following transfection with control siRNA or Sig1R siRNA, the ECs were stimulated with LPS, and cytoplasmic extracts were analyzed via immunoblotting for IκBα degradation. Sig1R depletion augmented LPS-induced IκBα degradation (Figure 6A,B), indicating that the potentiated nuclear uptake and subsequent DNA binding of NF-κB in Sig1R-depleted cells was secondary to the increased degradation of IκBα in the cytosol. This observation was further validated using an opposing approach. Contrary to Sig1R depletion, ECs pretreated with Sig1R activator PRE-084 reduced LPS-induced IκBα degradation (Figure 6C,D) in cytoplasmic extracts. Unlike PRE-084, Sig1R antagonist NE-100 had no effect on LPS-induced IκBα degradation (Figure 6E,F). Together, these data indicate that Sig1R limits EC inflammation via its ability to suppress the NF-κB signaling pathway. 

### 3.4. Sigma1R Activation Secondary to BiP/GRP78 Depletion/Inactivation Attenuates LPS-Induced Inflammatory Responses in EC 

Since the anti-inflammatory activity of Sig1R is contingent upon its dissociation from BiP/GRP78 at the MAM, we used knockdown BiP/GRP78 as another approach for the dissociation/activation of Sig1R and monitored the status of EC inflammation in BiP/GRP78-depleted cells. Consistent with the effect of PRE-084 (Figure 2A), the activation of Sig1R secondary to siRNA-mediated BiP/GRP78 depletion markedly reduced the expression of VCAM-1 caused by LPS (Figure 7A,B). Similarly, pretreatment of EC with Subtilase AB (SubAB), the prototype of a family of AB5 cytotoxins produced by Shiga toxigenic *Escherichia coli*, which is known to specifically cleave and inactivate BiP/GRP78, also showed a significant decrease in LPS-induced VCAM-1 expression (Figure 7C,D). These findings support the idea that the dissociation/activation of Sig1R, either through the depletion or inactivation of BiP/GRP78, is a critical requirement for its anti-inflammatory effect in ECs. 

To confirm the above possibility, we determined the effect of BiP/GRP78 depletion or inactivation on LPS-induced inflammatory responses in control versus Sig1R-depleted ECs. ECs were transfected with either control siRNA, Sig1R siRNA or BiP/GRP78 siRNA alone or in combination. A significant depletion of BiP/GRP78 and Sig1R was observed irrespective of whether the cells were transfected with siBiP/GRP78 or siSig1R alone or in combination (Figure 8A,C,D). Importantly, the depletion of BiP/GRP78 failed to block LPS-induced ICAM-1 expression in cells that were depleted of Sig1R compared to cells that were depleted of BiP/GRP78 alone. These results confirm that the protective effect of BiP/GRP78 depletion or inactivation on LPS-induced EC inflammation is mediated via Sig1R (Figure 8A,B). Next, we monitored the effect of BiP/GRP78 inactivation using SubAB on LPS-induced inflammatory response in Sig1R-depleted cells. Similar to BiP depletion, the inhibitory effect of BiP inactivation on LPS-induced ICAM-1 expression was blocked in cells that were depleted of Sig1R (Figure 9A,B). Together, these data establish that the attenuation of LPS-induced EC inflammation following the loss/inactivation of BiP/GRP78 is mediated by Sig1R. 

In order to further substantiate the above observation, we monitored the combined effect of Sig1R antagonist NE-100 and BiP/GRP78 antagonist SubAB on LPS-induced EC inflammation. As shown, the pretreatment of ECs with NE-100 failed to have an effect on LPS-induced VCAM-1 expression (Figure 4A–C), whereas the pretreatment of ECs with SubAB decreased LPS-induced VCAM-1 expression (Figure 7C,D). Both these observations are in line with the mode of action of NE-100 and SubAB, respectively. Interestingly, when ECs were pretreated with SubAB in combination with NE-100, we observed a significant decrease in LPS-induced VCAM-1 expression (Figure 10A,B), suggesting that the cleavage/inactivation of BiP/GRP78 by SubAB was enough to dissociate/activate Sig1R from BiP/GRP78 and thus overrides the effect of NE-100, which is to block the dissociation between Sig1R and BiP. These findings further substantiate the notion that the dissociation of Sig1R from BiP/GRP78 is a critical determinant of the anti-inflammatory action of Sig1R. 

### 3.5. Sigma1R Is Protective against Thrombin-Induced EC Permeability

In addition to endothelial inflammation, the loss of endothelial barrier integrity is a major pathogenic feature of ALI. Endothelial barrier integrity is mainly regulated by adherence junction proteins, such as VE-cadherin, and also through proteins that regulate actin cytoskeletal dynamics, such as the RhoA-actin pathway. Studies have shown that exposure of ECs to thrombin causes rearrangement in the actin cytoskeleton and promotes VE–cadherin disassembly, resulting in endothelial barrier disruption and increased EC permeability [48]. Furthermore, thrombin acts on ECs by cleaving its receptor PAR1, causing an increase in intracellular Ca^2+^ concentration and PKC activation, which, in turn, disrupts the VE–cadherin/catenin complex, leading to intercellular gap formation [41]. These findings prompted us to investigate the role of Sigma1R in regulating thrombin-induced changes in the actin cytoskeleton. We analyzed the effect of Sig1R depletion or activation on endothelial permeability using a trans-well in vitro permeability assay. ECs transfected with Sig1R siRNA or control siRNA were seeded onto trans-well inserts containing 1 µm pores within a transparent polyethylene terephthalate (PET) membrane coated with type-1 rat tail collagen. The confluent monolayer was treated with thrombin for 30 min, followed by the addition of high molecular weight FITC–Dextran on top of the cells. The movement of FITC–Dextran across the monolayer into the receiver tray is a direct measure of EC permeability or barrier disruption. Our results show that the depletion of Sig1R significantly potentiated thrombin-induced EC permeability, as indicated by a marked increase in fluorescent counts in the receiver tray (Figure 11A). In contrast, pretreatment with Sig1R activator PRE-084 significantly decreased the movement of FITC–Dextran across the monolayer (Figure 11B), thereby protecting EC barrier integrity. Next, we exposed the EC monolayer to plasma from septic patients and monitored its effect on endothelial permeability using trans-endothelial resistance (TER), which is a real-time measurement of EC permeability. We observed that the exposure of ECs to septic human plasma induced barrier disruption, as seen by a drop in TER. However, pretreatment with Sig1R activator PRE-084 protected barrier disruption caused by septic human plasma (Figure 11C). Together, our data show the central role of Sig1R activation in preserving EC barrier integrity. 

### 3.6. Sigma1R Regulates Thrombin-Induced Ca^2+^ Signaling in ECs 

To further dissect the mechanism of thrombin-induced EC permeability, we monitored the role of Sig1R in regulating Ca^2+^ signaling, a major determinant of EC permeability. ECs treated with PRE-084 or NE-100 were grown to confluence on 35 mm gelatin-coated glass coverslips. A cell-permeable fluorescent probe Flou-4AM was added to the confluent monolayer to measure changes in cytosolic Ca^2+^. Following exposure to Flou-4AM for 25 min at 37C, the cells were gently washed with Ca^2+^-free HBSS buffer, and coverslips were mounted on a confocal microscope. Ca^2+^ release from the ER stores was measured by perfusing Ca^2+^-free imaging buffer and stimulating cells with thrombin. The results showed that Sig1R activation by PRE-084 significantly blocked thrombin-induced Ca^2+^ release from the ER stores, whereas Sig1R antagonist NE-100 had no effect on thrombin-induced Ca^2+^ release from the intracellular ER stores (Figure 12). These data underscore the importance of Sig1R in controlling EC permeability via its ability to regulate Ca^2+^ signaling. 

### 3.7. Sigma1R Activation Protects against Sepsis-Induced Lung Inflammation and Injury 

We next validated our in vitro findings by using an in vivo mouse model of sepsis-induced ALI. We used the cecal ligation and puncture (CLP) method, a widely used procedure for mimicking sepsis in vivo. C57BL/6 mice were injected intraperitoneally with Sig1R agonist PRE-084 (0.5 mg/kg) or saline 1 h prior to CLP or SHAM (ligation but no puncture) surgery to induce sepsis. Twelve hours after CLP, the lungs were analyzed for markers of inflammation and injury. We observed that compared to SHAM, the lungs from CLP mice showed a significant increase in ICAM-1 and IL-6 levels, and these responses were strongly inhibited in CLP mice treated with PRE-084 (Figure 13A,B). Consistent with this, infiltration of neutrophils in the lungs, as measured in terms of myeloperoxidase (MPO) activity [49], was also blocked in CLP mice treated with PRE-084 (Figure 13C). Next, we determined the effect of PRE-084 on lung tissue edema and vascular leak in the mice by measuring the lung wet-to-dry ratio and pulmonary extravasation of Evan’s blue dye (EBD). We observed that both tissue edema and lung vascular leak were protected in the lungs of CLP mice treated with PRE-084 (Figure 13D,E). Together, these data establish the protective role of Sig1R in sepsis-induced lung vascular inflammation and injury. 

## 4. Discussion

MAM represents a nexus for many signaling cascades and biochemical reactions [8,10]. Perturbation in ER–mitochondria interface/axis is linked to the pathogenesis of many diseases, in particular, metabolic and neurodegenerative disorders [15,18]. However, the importance of MAM proteins in regulating EC function in the context of ALI is under explored. Here, we provide evidence that Sig1R plays a central role in protecting against EC inflammation and permeability and is a druggable target against ALI in sepsis. 

The role of Sig1R in neuronal and cardiac cells has been well documented; however, its relevance in relation to endothelial cells has received little attention. We investigated the role and regulation of Sig1R in the context of pulmonary endothelium, the primary target of pulmonary and extrapulmonary insults associated with lung inflammatory diseases, such as ALI. To this end, we used two opposing approaches to address the relevance of Sig1R in EC inflammation. In the first approach, endothelial cells were depleted of Sig1R using siRNA-mediated knockdown, and in the second approach, endogenous Sig1R was activated using PRE-084, which activates Sig1R by causing its dissociation from BiP/GRP78 (Figure 1 and Figure 2). We observed that ECs lacking Sig1R, when exposed to LPS, displayed augmented inflammatory response. In contrast, ECs exposed to Sig1R agonist PRE-084 showed an attenuated LPS-induced inflammatory response, whereas the Sig1R antagonist NE-100, which inactivates Sig1R by blocking its dissociation with BiP/GRP78, had no effect on LPS-induced EC inflammation. Together, these data reveal the protective role of Sig1R against EC inflammation. Not to overlook the non-specific effects of PRE-084 toward Sig1R, we monitored the effect of PRE-084 in cells that were depleted of Sig1R. PRE-084 failed to exert its protective effect on LPS-induced inflammation in cells lacking Sig1R, validating its specificity toward Sig1R. Similar to LPS, the activation of Sig1R by PRE-084 significantly mitigated EC inflammation caused by thrombin, another important proinflammatory and edemagenic mediator whose levels are elevated in plasma and bronchoalveolar lavage (BAL) fluids of patients suffering from ALI [50]. These findings indicate that Sig1R manifests its anti-inflammatory activity irrespective of the stimulus used and may represent a common mechanism in terms of limiting EC inflammation.

We next investigated the mechanism by which Sig1R exerts its anti-inflammatory function in ECs. To this end, we examined the role of Sig1R in regulating the NF-κB canonical pathway (Figure 14). Our data revealed that cells lacking Sig1R exhibited increased LPS-induced IκBα degradation, whereas ECs pretreated with Sig1R agonist PRE-084 showed decreased LPS-induced IκBα degradation. In contrast, Sig1R antagonist NE-100 has no effect on LPS-induced IκBα degradation. In accordance with the classical NF-κB signaling pathway, Sig1R-depleted cells showed increased RelA/p65 (NF-kB) nuclear translocation and subsequent DNA binding, resulting in increased proinflammatory gene expression (ICAM-1, VCAM-1, IL-6, and IL-8). Unlike ECs, studies in human embryonic kidney (HEK293) cells and bone marrow-derived macrophages (BMDMs) have shown that Sig1R interacts with IRE1α and restricts its LPS-induced endonuclease activity and subsequently blocks inflammatory cytokine production without involving the NF-κB, JNK and ERK inflammatory pathways [51]. Furthermore, the depletion of Sig1R in motor neurons led to cell death, whereas no defect was observed in sensory neurons upon Sig1R knockdown. Together, these findings evidence the importance of the cellular context in influencing the action and the signaling cascades activated by Sig1R.

Studies by Hayashi et al. have shown that in CHO cells, the endogenous Sig1R co-immunoprecipitated exclusively with endogenous BiP/GRP78 but not with other ER chaperones, indicating a highly specific interaction between them. Importantly, the functionality of Sig1R has been shown to be dependent on its interaction with BiP/GRP78 [34]. Earlier work from our lab demonstrated that the depletion or inactivation of BiP/GRP78 mitigates inflammatory signaling both in primary endothelial cell cultures and in a mouse model of ALI. These observations prompted us to hypothesize that the mechanism behind BiP/GRP78 depletion/inactivation-mediated mitigation of EC inflammation involves dissociation and, thereby, the activation of Sig1R from BiP/GRP78. Our hypothesis was validated by the findings that siRNA-mediated BiP/GRP78 depletion or SubAB-mediated BiP/GRP78 inactivation each failed to mitigate LPS-induced EC inflammation in cells that were depleted of Sig1R. Furthermore, HPAECs pretreated with NE-100 in combination with SubAB when exposed to LPS were inhibited in terms of inflammatory responses, indicating that the cleavage/inactivation of BiP/GRP78 by SubAB was sufficient for Sig1R to be liberated and exert its anti-inflammatory effects, thereby overriding the action of NE-100. Altogether, our in vitro data establish a protective role of Sig1R in regulating EC inflammation.

In addition to inflammation, a key component of ALI in sepsis is the disruption of the vascular endothelium barrier, leading to increased microvascular permeability and generation of protein-rich pulmonary edema. Our studies showed that disabling Sigma1R augmented EC permeability, whereas activating Sigma1R protected against EC permeability in response to thrombin. Our data were further substantiated when EC barrier disruption caused by human septic plasma was protected in the presence of PRE-084. These observations are in line with earlier studies in human dermal lymphatic endothelial cells (HDLECs) and human umbilical vein endothelial cells (HUVECs), where Sig1R is shown to be protective against endothelial barrier disruption through its ability to enhance glycolytic energy production [52,53]. Furthermore, since Ca^2+^ plays a critical role in intracellular signaling and is a critical mediator of EC permeability, we also tested the ability of Sigma1R to control Ca^2+^ signaling in EC. Our data showed that thrombin-induced Ca^2+^ release from the ER was attenuated upon the activation of Sig1R by PRE-084, whereas the inactivation of Sig1R showed no effect on the above response. Thus, the Ca^2+^ released from the ER, in turn, activates downstream signaling to engage stromal-interaction molecule 1 (STIM1), an ER-localized Ca^2+^ sensor protein, Orai1 or TRPC (transient receptor potential canonical) channels, and thereby promotes Ca^2+^ entry. Studies have associated Sig1R with Ca^2+^ signaling, both at the level of Ca^2+^ release from ER and Ca^2+^ entry through the regulation of ion channels [54]. However, it remains to be addressed whether Sig1R mediates Ca^2+^ signaling in ECs via IP3R and STIM1 (STIM-Orai or STIM-TRPC1/4/6).

## 5. Conclusions

In summary, our integrated in vitro and in vivo studies identify the critical role of Sig1R in limiting EC inflammation and permeability associated with ALI and support the idea that targeting Sig1R for its activation could prove a viable therapeutic strategy against ALI in sepsis.

## Figures and Tables

**Figure 1 cells-13-00005-f001:**
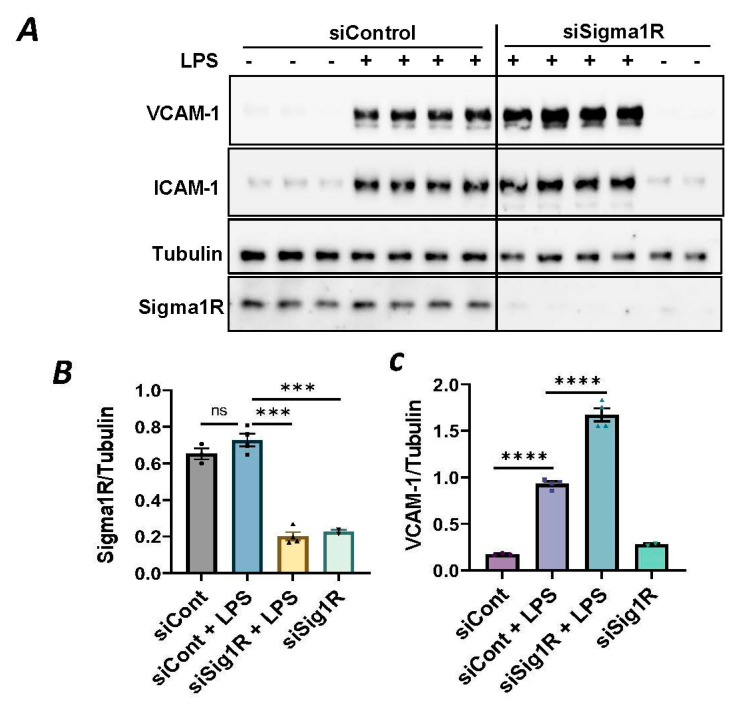
siRNA-mediated depletion of Sigma1R potentiated LPS-induced inflammatory responses. HPAECs were transfected with control siRNA or Sigma1R siRNA for 48 h followed by LPS treatment (1 μg/mL) for 6 h. (**A**) Total cell lysate was immunoblotted for VCAM-1 and ICAM-1 using anti-ICAM-1 and anti-VCAM-1 antibody. Anti-tubulin antibody was used as a loading control. Anti-Sigma1R antibody was used to monitor Sigma1R depletion. (**B**,**C**) Bar graphs represents the effect of Sigma1R knockdown on LPS-induced VCAM-1 and ICAM-1 normalized to tubulin levels. (**D**) Bar graph represents the effect of Sigma1R knockdown in Sigma1R normalized to tubulin levels. (**E**) Cell culture supernatants were analyzed for IL-8 using ELISA. Data was analyzed using one-way ANOVA followed by post hoc Tukey’s test. Data are mean ± S.E. (*n* = 3–4 per condition; *ns*-not significant, *p* > 0.05, * *p* < 0.05, ** *p* < 0.01, *** *p* < 0.001, **** *p* < 0.0001).

**Figure 2 cells-13-00005-f002:**
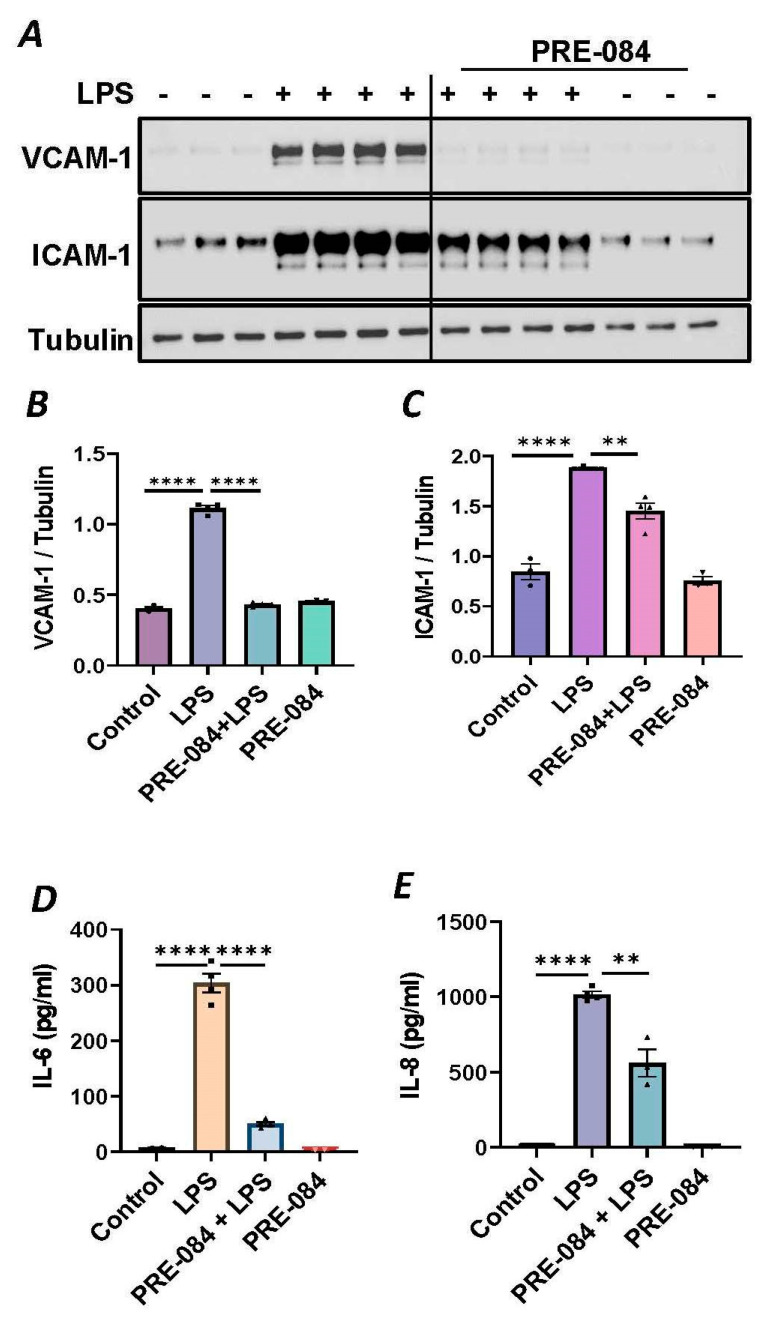
Activation of Sigma1R-inhibited LPS or thrombin-induced inflammatory responses. HPAECs were pretreated with Sigma1R agonist PRE-084 (10 µM) for 1 h followed by LPS treatment (1 μg/mL) for 6 h. (**A**) Total cell lysate was immunoblotted for VCAM-1 and ICAM-1 using anti-ICAM-1 and anti-VCAM-1 antibody. Anti-tubulin antibody was used as a loading control. (**B**,**C**) Bar graphs represents the effect of PRE-084 on LPS-induced VCAM-1 and ICAM-1 normalized to tubulin levels. (**D**,**E**) Cell culture supernatants were analyzed for IL-6 and IL-8 using ELISA. (**F**) HPAECs were pretreated with Sigma1R agonist PRE-084 (10 µM) for 1 h followed by thrombin treatment (5 U/mL) for 6 h. Total cell lysate was immunoblotted for VCAM-1 using anti-VCAM-1 antibody. Anti-β-actin antibody was used as a loading control. (**G**) Bar graph represents the effect of PRE-084 on thrombin-induced VCAM-1 normalized to actin levels. Data was analyzed using one-way ANOVA followed by post hoc Tukey’s test. Data are mean ± S.E. (*n* = 3–4 per condition; ** *p* < 0.01, *** *p* < 0.001, **** *p* < 0.0001). (**H**) Cell culture supernatants were analyzed for IL-8 using ELISA. Data was analyzed using one-way ANOVA followed by post hoc Tukey’s test. Data are mean ± S.E. (*n* = 3–9 per condition; **** *p* < 0.0001).

**Figure 3 cells-13-00005-f003:**
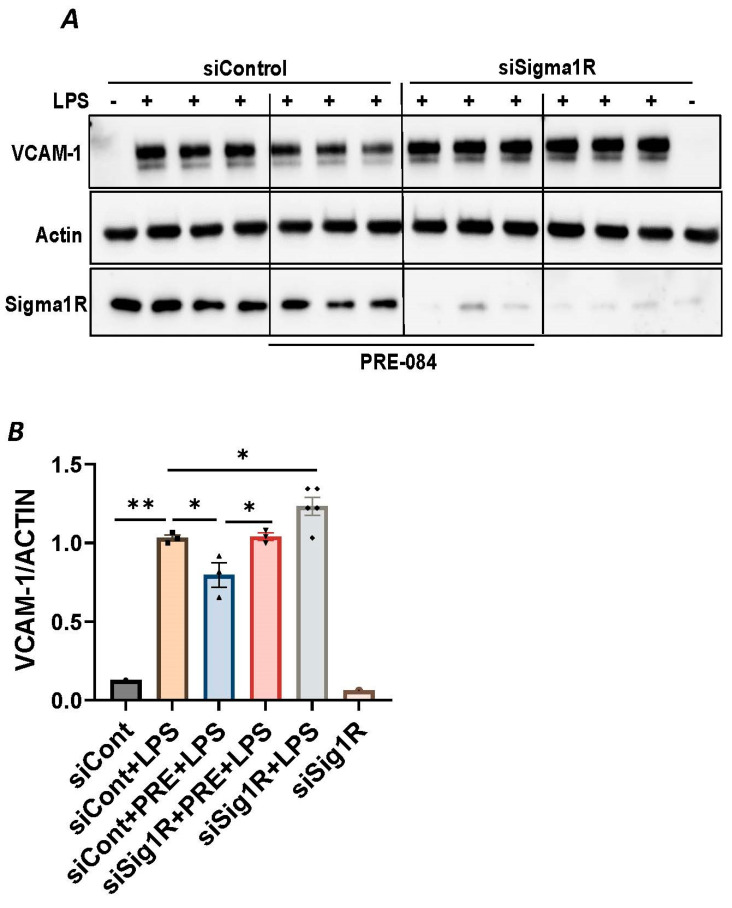
PRE-084 failed to inhibit LPS-induced inflammatory response in Sigma1R-depleted cells. HPAECs were transfected with control siRNA or Sigma1R siRNA. 48 h later cells were pretreated with PRE-084 (20 µM) for 1 h followed by LPS treatment (1 μg/mL) for 6 h. (**A**) Total cell lysates were immunoblotted for VCAM-1 using anti-VCAM-1 antibody and β-Actin was monitored as loading control. (**B**) Bar graph represents the effect of Sigma1R knockdown with and without PRE-084 on LPS-induced VCAM-1 normalized to actin levels. Data was analyzed using one-way ANOVA followed by post hoc Tukey’s test. Data are mean ± S.E. (*n* = 1–3 per condition; * *p* < 0.05, ** *p* < 0.01).

**Figure 4 cells-13-00005-f004:**
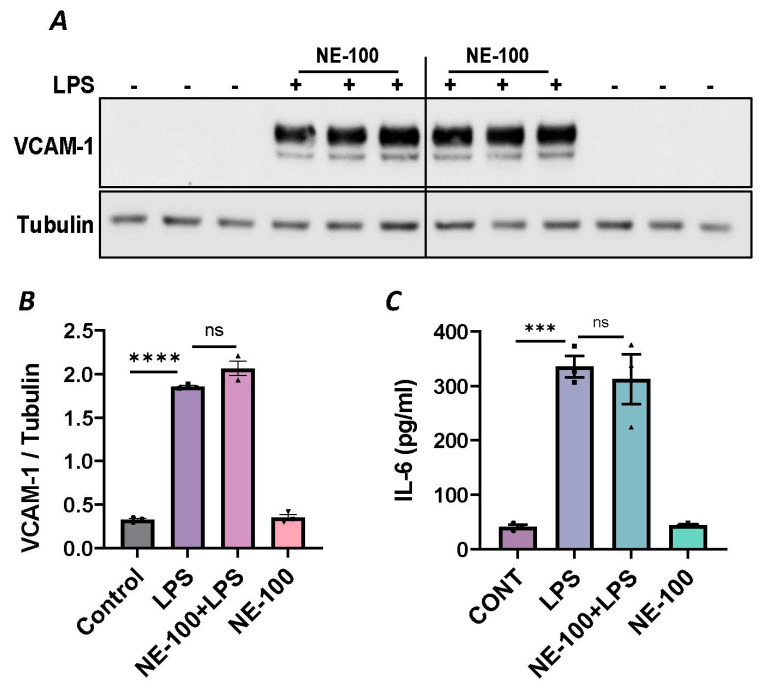
Sigma1R antagonist NE-100 has no effect on LPS-induced inflammatory responses. HPAECs were pretreated with Sigma1R antagonist NE-100 (20 µM) for 1 h followed by LPS treatment (1 μg/mL) for 6 h. (**A**) Total cell lysates were immunoblotted for VCAM-1 using anti-VCAM-1 antibody and tubulin was monitored as loading control. (**B**) Bar graph represents the effect of NE-100 on LPS-induced VCAM-1 normalized to tubulin levels. (**C**) Cell culture supernatants were analyzed for IL-6 using ELISA. Data was analyzed using one-way ANOVA followed by post hoc Tukey’s test. Data are mean ± S.E. (*n* = 3 per condition; *** *p* < 0.001, **** *p* < 0.0001, *ns*-not significant).

**Figure 5 cells-13-00005-f005:**
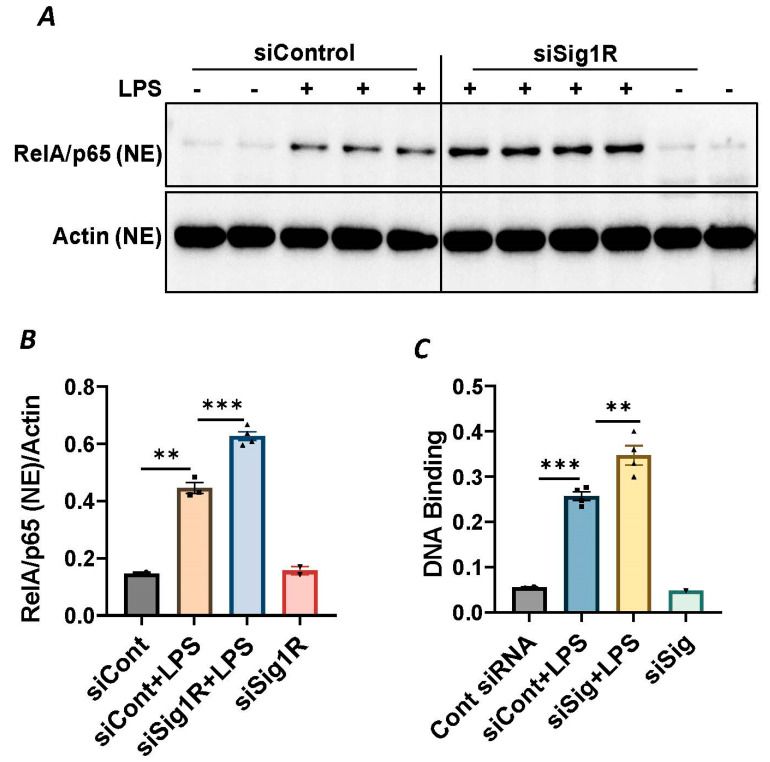
Sigma1R depletion potentiated LPS-induced RelA/p65 nuclear translocation and subsequent DNA binding. HPAECs were transfected with control siRNA or Sigma1R siRNA. After 48 h, cells were challenged with LPS (1 μg/mL) for 1 h. (**A**) Nuclear extracts were separated by SDS-PAGE and immunoblotted for RelA/p65 using anti-RelA/p65 antibody and for β-Actin using anti- β-Actin antibody (**B**) Bar graph represents the effect of Sigma1R knockdown on LPS-induced nuclear translocation of RelA/p65 normalized to actin levels. Data was analyzed using one-way ANOVA followed by post hoc Tukey’s test. Data are mean ± S.E. (*n* = 2–4 per condition; ** *p* < 0.01, *** *p* < 0.001). (**C**) HPAECs were transfected with control siRNA or Sigma1R siRNA. After 48 h, cells were challenged with LPS (1 μg/mL) for 1 h. Nuclear extracts were prepared and assayed for DNA binding of RelA/p65 using Cayman’s NF-κB (RelA/p65) Transcription Factor Assay Kit as described in Materials and Methods. Data was analyzed using one-way ANOVA followed by post hoc Tukey’s test. Data are mean ± S.E. (*n* = 1–4 per condition; ** *p* < 0.01, *** *p* < 0.001).

**Figure 6 cells-13-00005-f006:**
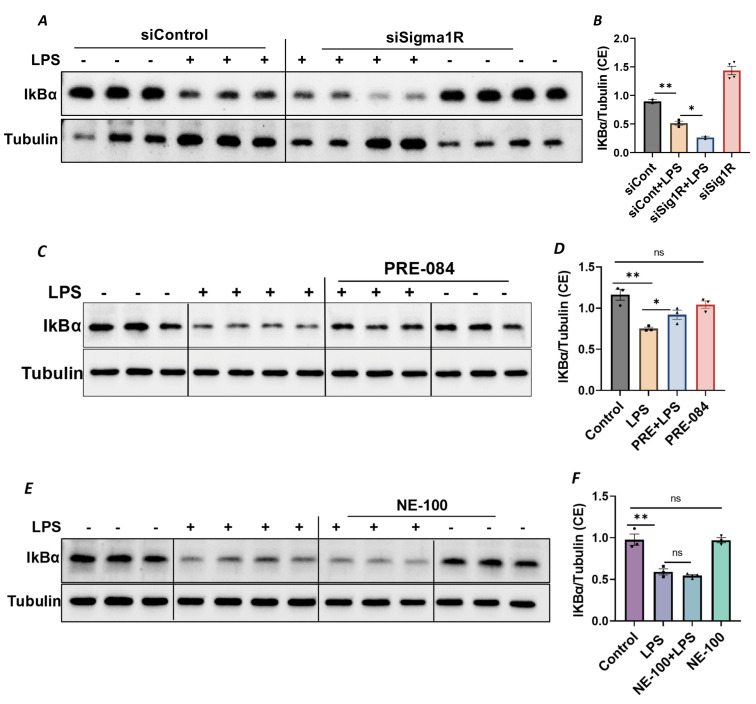
Sigma1R depletion promoted LPS-induced IκBα degradation, whereas Sigma1R activation reduced LPS-induced IκBα degradation. HPAECs were transfected with control siRNA or Sigma1R siRNA. After 48 h, cells were challenged with LPS (1 μg/mL) for 1 h. (**A**) Cytoplasmic extracts were separated by SDS-PAGE and immunoblotted for IκBα using anti-IκBα antibody and for tubulin using anti-tubulin antibody (**B**) Bar graph represents the effect of Sigma1R knockdown on LPS-induced IκBα degradation normalized to tubulin levels. (**C**) HPAECs were pretreated with Sigma1R agonist PRE-084 (10 µM) for 1 h followed by LPS treatment (1 μg/mL) for 6 h. Cytoplasmic extracts were separated by SDS-PAGE and immunoblotted for IκBα using anti-IκBα antibody and for tubulin using anti-tubulin antibody (**D**) Bar graph represents the effect of Sigma1R activation on LPS-induced IκBα degradation normalized to tubulin levels. (**E**) HPAECs were pretreated with Sigma1R antagonist NE-100 (10 µM) for 1 h followed by LPS treatment (1 μg/mL) for 6 h. Cytoplasmic extracts were separated by SDS-PAGE and immunoblotted for IκBα using anti-IκBα antibody and for tubulin using anti-tubulin antibody. (**F**) Bar graph represents the effect of Sigma1R inactivation on LPS-induced IκBα degradation normalized to tubulin levels. Data was analyzed using one-way ANOVA followed by post hoc Tukey’s test. Data are mean ± S.E. (*n* = 2–4 per condition; *ns*-not significant, *p* > 0.05, * *p* < 0.05, ** *p* < 0.01).

**Figure 7 cells-13-00005-f007:**
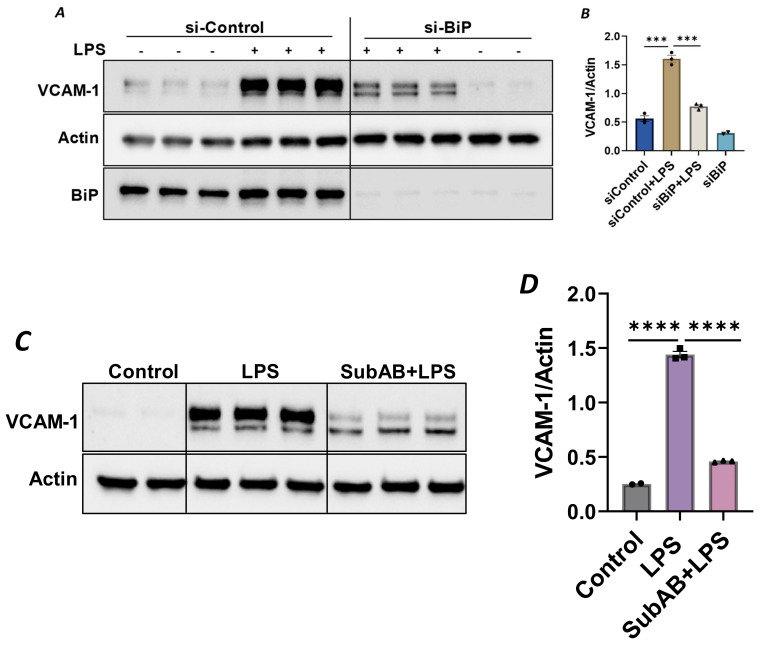
Depletion or inactivation of BiP/GRP78 inhibited LPS-induced VCAM-1 protein expression. HPAECs were transfected with control siRNA or BiP siRNA for 48 h, followed by LPS treatment (1 μg/mL) for 6 h. (**A**) Total cell lysate was immunoblotted for VCAM-1 using anti-VCAM-1 antibody. Anti-actin antibody was used as a loading control. Anti-BiP/GRP78 antibody was used to monitor BiP/GRP78 depletion. (**B**) Bar graph represents the effect of BiP/GRP78 knockdown on LPS-induced VCAM-1 normalized to actin levels. (**C**) HPAECs were pretreated with BiP inhibitor SubAB for 3 h, followed by LPS treatment (1 μg/mL) for 6 h. Total cell lysate was immunoblotted for VCAM-1 using anti-VCAM-1 antibody. Anti-actin antibody was used as a loading control. (**D**) Bar graph represents the effect of BiP/GRP78 inactivation via SubAB on LPS-induced VCAM-1 normalized to actin levels. Data were analyzed using one-way ANOVA, followed by post hoc Tukey’s test. Data are mean ± S.E. (*n* = 2–3 per condition; *** *p* < 0.001, **** *p* < 0.0001).

**Figure 8 cells-13-00005-f008:**
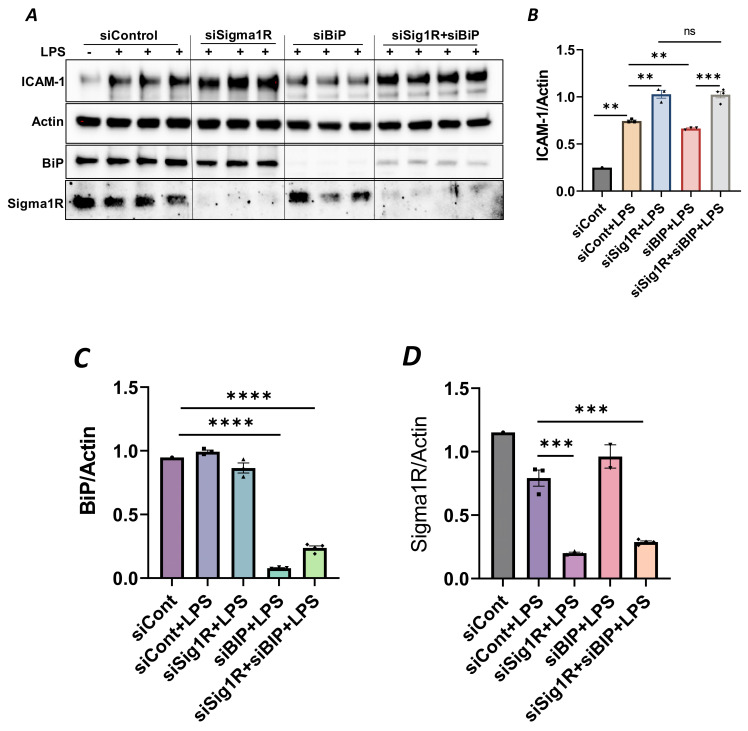
Attenuation of LPS-induced EC inflammation upon BiP/GRP78 depletion is mediated via Sigma1R. HPAECs were transfected with control siRNA or BiP siRNA or siSigma1R, alone or in combination for 48 h, followed by LPS treatment (1 μg/mL) for 6 h. (**A**) Total cell lysate was immunoblotted for ICAM-1 using anti-ICAM-1 antibody. Anti-actin antibody was used as a loading control. Anti-BiP/GRP78 and anti-Sigma1R antibodies were used to monitor BiP/GRP78 and Sigma1R depletion, respectively. (**B**) Bar graph represents the effect of BiP/GRP78 or Sigma1R depletion alone or in combination on LPS-induced ICAM-1 normalized to actin levels. (**C**,**D**) Bar graphs represent the effect of BiP/GRP78 or Sigma1R depletion alone or in combination on levels of BiP/GRP78 or Sigma1R normalized to actin levels. Data were analyzed using one-way ANOVA followed by post hoc Tukey’s test. Data are mean ± S.E. (*n* = 1–4 per condition; *ns*-not significant, *p* > 0.05, ** *p* < 0.01, *** *p* < 0.001, **** *p* < 0.0001).

**Figure 9 cells-13-00005-f009:**
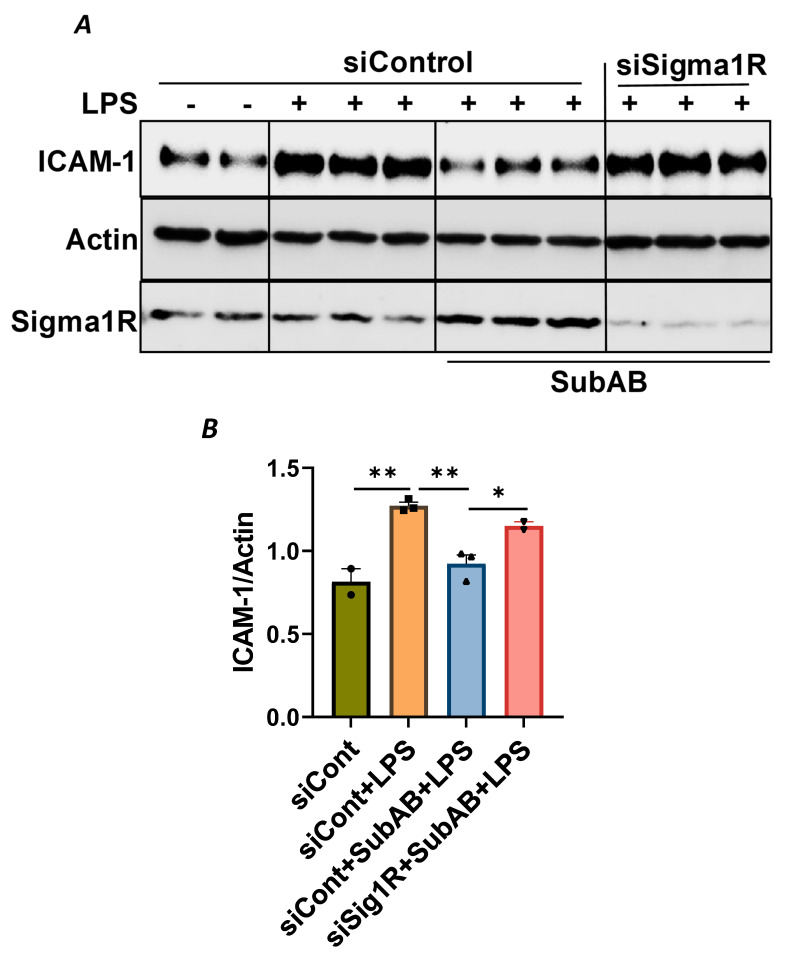
Attenuation of LPS-induced EC inflammation following BiP/GRP78 inactivation by SubAB is mediated by Sigma1R. HPAECs were transfected with control siRNA or Sigma1R siRNA. After 48 h, cells were pretreated with SubAB for 3 h, followed by LPS treatment (1 μg/mL) for 6 h. (**A**) Total cell lysates were immunoblotted for ICAM-1 using anti-ICAM-1 antibody and β-Actin was monitored as loading control. Anti-Sigma1R antibody was used to monitor Sigma1R depletion. (**B**) Bar graph represents the effect of Sigma1R depletion alone or in combination with SubAB on LPS-induced ICAM-1 normalized to actin levels. Data were analyzed using one-way ANOVA followed by post hoc Tukey’s test. Data are mean + S.E. (*n* = 2–3 per condition; * *p* < 0.05, ** *p* < 0.01).

**Figure 10 cells-13-00005-f010:**
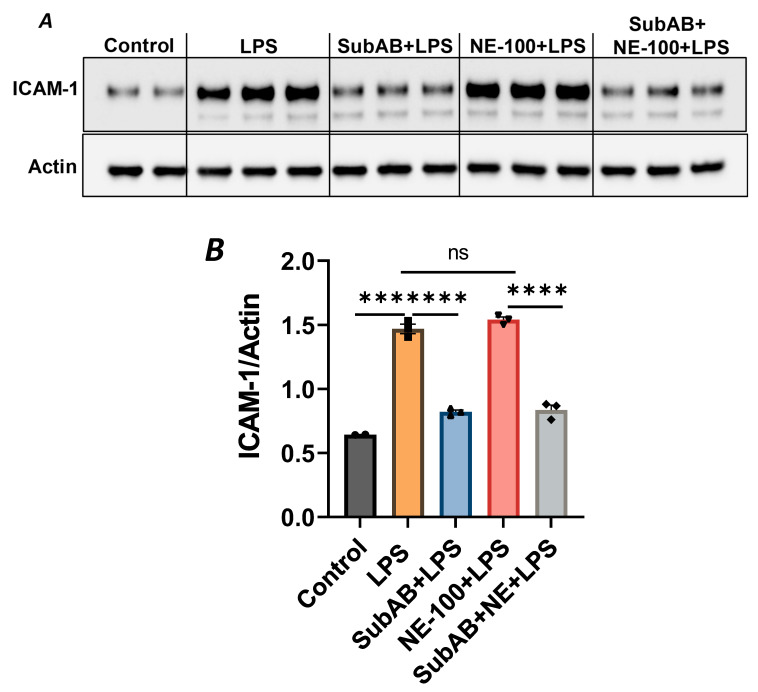
BiP/GRP78 inactivator SubAB overrides the effect of Sigma1R antagonist NE-100. HPAECs were treated with BiP inactivator SubAB or Sigma1R antagonist NE-100, alone or in combination for 3 h, followed by LPS treatment (1 μg/mL) for 6 h. (**A**) Total cell lysates were immunoblotted for ICAM-1 using anti-ICAM-1 antibody, and actin was monitored as loading control. (**B**) Bar graph represents the effect of NE-100 or SubAB alone or in combination on LPS-induced ICAM-1 normalized to actin levels. Data were analyzed using one-way ANOVA, followed by post hoc Tukey’s test. Data are mean + S.E. (*n* = 2–3 per condition; *** *p* < 0.001, **** *p* < 0.0001, *ns*-not significant).

**Figure 11 cells-13-00005-f011:**
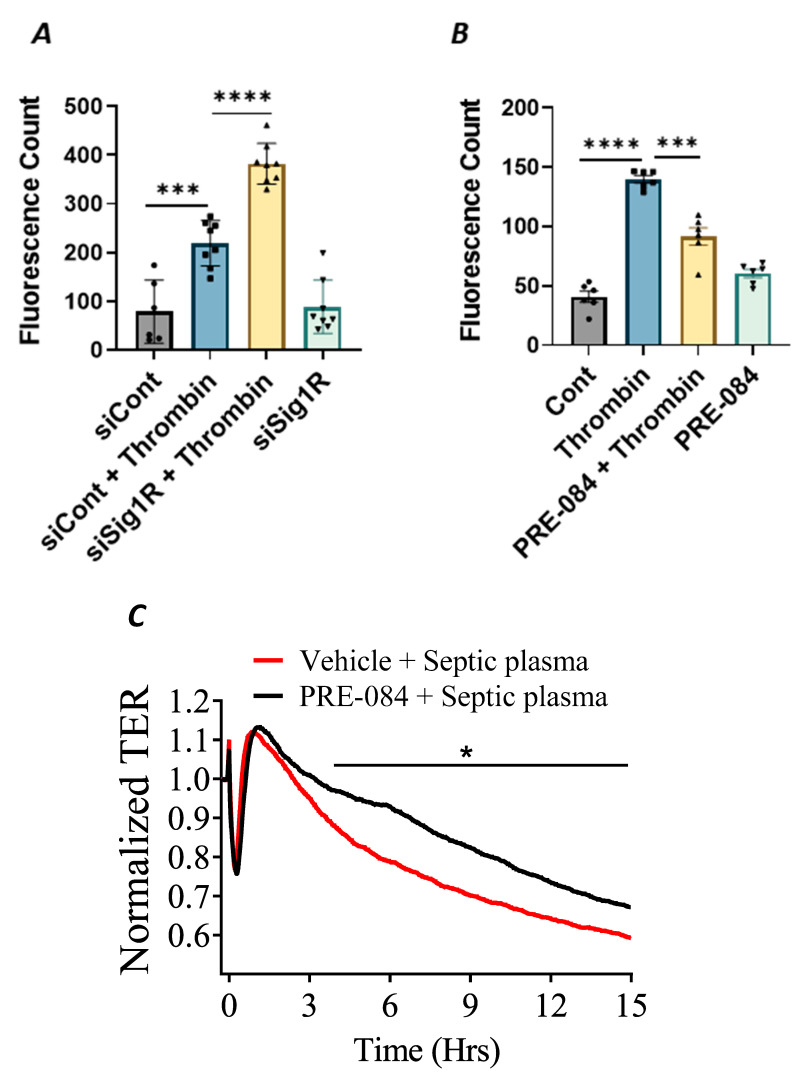
Activation of Sigma1R protected against thrombin-induced EC permeability. HPAECs transfected with siControl or siSig1R were seeded at 20,000 cells per trans-well insert and cultured for 48 h (**A**). (**B**) HPAECs were treated with PRE-084 (10 µM) for 1 h. Following the treatment (**A**,**B**), the confluent cell monolayer was challenged with thrombin (5 U/mL) for thirty minutes. Monolayer integrity was determined using the FITC–Dextran permeability test. Permeation of the dye was halted once the inserts were removed from the wells. To measure fluorescence intensity, media was transferred from receiver tray to an opaque 96-well plate. A fluorescent plate reader equipped with filters for 485 nm and 585 nm excitation and emission was used to quantify FITC fluorescence intensity (shown as fluorescence count). Data were analyzed using one-way ANOVA followed by post hoc Tukey’s test. The data are presented as means ± S.E. (*n* = 6–8 for each condition; *** and **** *p* < 0.05). (**C**) Sigma1R activation protected barrier disruption caused by septic human plasma. Real-time TER changes in HPAECs treated with 0.1% plasma from patients with sepsis with and without Sigma1R agonist PRE-084 were monitored for 15 h. The TER was normalized to baseline (*n* = 4 per condition, * *p* < 0.05). Data were analyzed using two-tailed, unpaired Student’s *t* test.

**Figure 12 cells-13-00005-f012:**
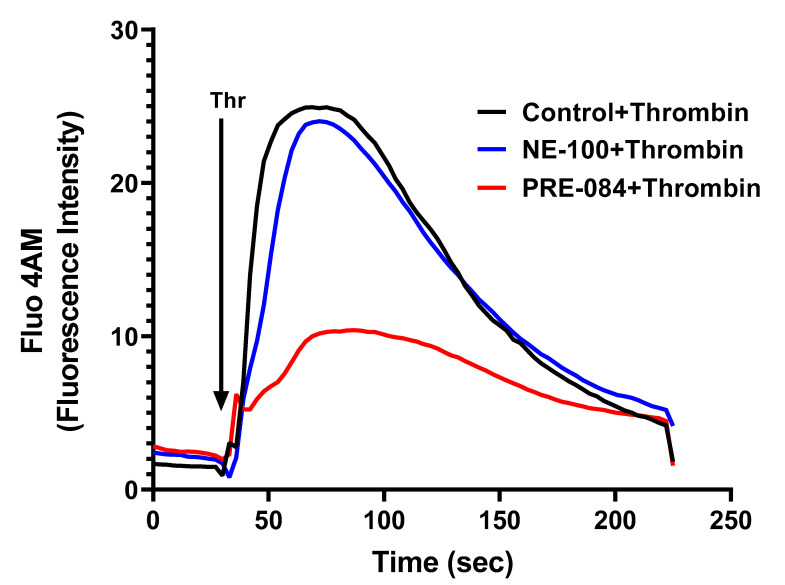
Sigma1R activation by PRE-084-blocked thrombin-induced Ca^2+^ release from the ER stores. HPAECs grown to confluence on 35 mm gelatin-coated coverslips were left untreated or treated with PRE-084 or NE-100 for 1 h at 37 °C. Following treatment, cells were loaded with Fluo-4AM (5 µM) for 25 min and then washed twice with Ca^2+^-free HBSS buffer and mounted on a confocal microscope. Cytosolic Ca^2+^ was measured in response to thrombin (2.5 U/mL) under extracellular depletion–repletion conditions. Mean fluorescence intensity (presented as Fluo-4 AM fluorescence intensity) was assessed from 110 to 120 cells per condition.

**Figure 13 cells-13-00005-f013:**
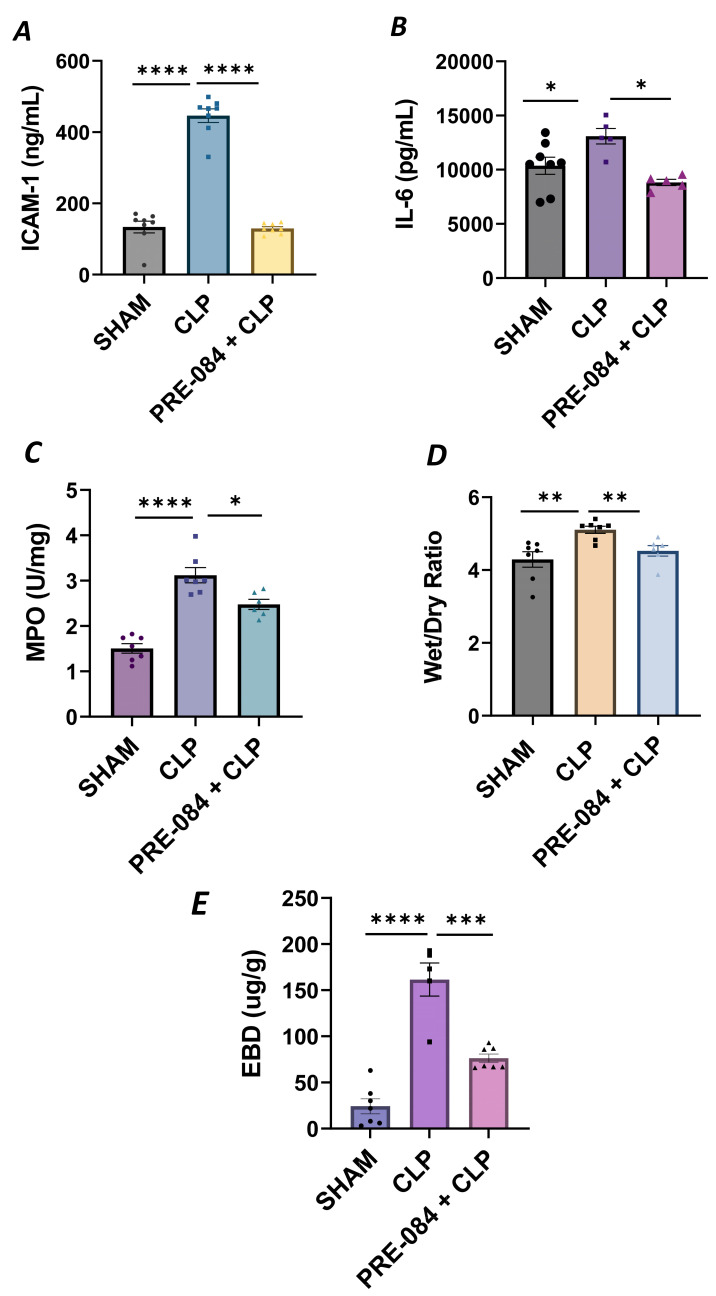
Sigma1R activation blocked sepsis-induced lung injury and inflammation. Wild-type C57BL/6L mice were injected intraperitoneally with PRE-084 (0.5 mg/kg) 1 h prior to SHAM or CLP. Twelve hours after CLP, lung homogenates were analyzed for proinflammatory mediators (**A**) ICAM-1 and (**B**) IL-6. Neutrophil infiltration was analyzed as a measure of (**C**) myeloperoxidase (MPO) activity. Vascular leak was analyzed as a measure of (**D**) wet-to-dry weight ratio and (**E**) Evan‘s blue extravasation (EBD). Data were analyzed using one-way ANOVA, followed by post hoc Tukey’s test. Data are presented as mean ± S.E. (*n* = 5–8 per condition; * *p* ≤ 0.05, ** *p* < 0.01, *** *p* < 0.001, **** *p* < 0.0001).

**Figure 14 cells-13-00005-f014:**
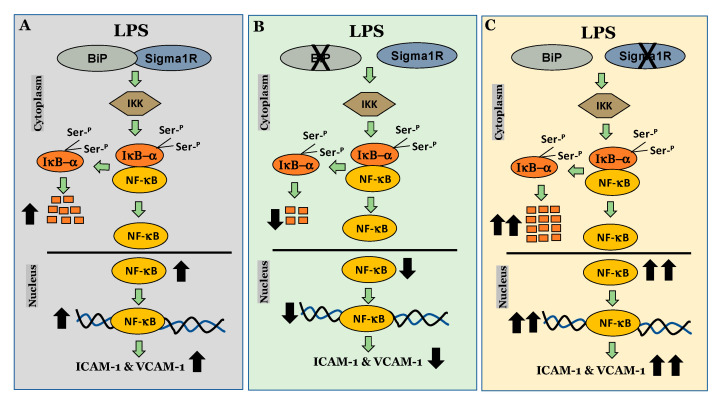
Schematic showing the anti-inflammatory role of Sigma1R in regulating NF-κB canonical pathway in endothelial cells. (**A**) inactivation of sigma1R via its association with BiP contributes to LPS-induced proinflammatory gene expression (ICAM-1 and VCAM-1) (**B**) sigma1R dissociation from BiP/GRP78 inhibits LPS-induced inflammation. (**C**) depletion of sigma1R further potentiates LPS-induced inflammation.

## Data Availability

Data are contained within the article.

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
