# Peer review of "The Protective Role of Mitochondria-Associated Endoplasmic Reticulum Membrane (MAM) Protein Sigma-1 Receptor in Regulating Endothelial Inflammation and Permeability Associated with Acute Lung Injury"

_cells, 2023, doi:10.3390/cells13010005_

Round 1
Reviewer 1 Report
Comments and Suggestions for Authors
The manuscript by Mahamed Z., et al titled “The Protective role of mitochondrial associated ER membrane (MAM) protein Sigma-1 receptor…….acute lung injury” investigates the role of Sigma-1R receptor (Sig1R), an endoplasmic reticulum receptor chaperone, in endothelial injury and dysfunction associated with acute lung injury (ALI). The authors used both in vitro and in vivo studies to address the hypothesis that the mechanism behind BiP/GRP78 depletion/inactivation-mediated mitigation of EC inflammation in ALI involves dissociation and thereby activation of Sig1R from BiP/GRP7.
To examine this hypothesis, the authors utilized various approaches, including pharmacological interventions and genetic knockdown techniques in primary endothelial cells derived from the human lung. Their findings demonstrated that the absence of Sig-1R exacerbates endothelial inflammation, increases permeability, and induces injury in both an LPS-induced injury model and a thrombin-induced injury model. Furthermore, the authors revealed that knocking down the endoplasmic reticulum chaperone BiP/GRP78 also provides protection against endothelial injury. Notably, this protective effect was observed to be dependent on the presence of Sig-1R. Importantly, the study extended its investigation to a well-established in vivo model of sepsis-induced ALI, confirming their results and highlighting the role of Sig1R in protecting against endothelial dysfunction associated with ALI.
The experiments in the study were meticulously conducted and included appropriate control groups for each set of experiments. Additionally, the research identifies Sig-1R as a potential target for therapy in the context of acute lung injury (ALI).
Author Response
Thank you very much for taking the time to review the manuscript. We appreciate that you find our study meticulously conducted and translationally relevant.
Reviewer 2 Report
Comments and Suggestions for Authors
The objective of this study by Mahamed and colleagues was to investigate the protective role of mitochondria associated endoplasmic reticulum membrane (MAM) protein Sigma-1 receptor in regulating endothelial inflammation and permeability associated with acute lung injury. Acute lung injury and the acute respiratory distress syndrome (ARDS) represent a spectrum of progressive respiratory failure affecting over 190,000 patients annually in the United States and causing 75,000 deaths. ARDS is a life-threatening lung injury with a 38.5% case fatality rate. Emerging evidence indicate that organelles within the cell engage in extensive communication either directly or indirectly through membrane contacts. Communication between endoplasmic reticulum (ER) and mitochondrion (MITO), two multifunctional organelles, is central to many cellular processes including Ca2+ homeostasis, inflammasome assembly, mitochondrial dynamics, ER stress, cell survival, and lipid metabolism. MAM represents a nexus for many signaling cascades and biochemical reactions. Perturbation in ER-mitochondria interface/axis is linked to the pathogenesis of many diseases, in particular, metabolic and neurodegenerative disorders. However, the importance of MAM proteins in regulating EC function in the context of ALI is not clear. In this article, authors provide evidence that Sig1R plays a central role in protecting against endothelial cell inflammation and permeability and is a druggable target against ALI in sepsis. The conceptual framework and the design of the experiments are straight forward and the discussion and conclusion are adequately supported by the experimental data. My minor concerns are listed below:
a) In the title, ER (line 2) should be written as Endoplasmic Reticulum
b) In the third paragraph of introduction, it is stated that Sigma1R ligands include psychotropic and neuroprotective agents and many of them are currently in clinical use. However, quoted references (references 29-32) does not support this statement.
c) It is not clear whether Triton X or Triton X - 100 (line 112) used in immunoblot analysis.
d) Is IL-8 a cytokine or chemokine (line 120)?
e) What is the name of the fluorescence plate reader used in in vitro permeability experiment?
f) Mouse lung contain 5 lobes. It is not clear which of these 5 lung lobes were used (see lines 190 - 192) to calculate lung wet-to-dry weight as well as for other experiments.
g) Section on the statistical analysis is not clearly written. Specific statistical analysis should be described in the accompanying figure legends.
Reviewer 3 Report
Comments and Suggestions for Authors
The authors in this study have carried out an in-depth analysis of a mitochondria associated ER membrane protein Sigma-1 receptor for its protective role in endothelial cells. They have utilized siRNA to knockdown Sig1R; pharmacological agent PRE-084 known to activate Sig1R by dissociating it from BiP; and NE-100 which is known to inactivate SigR1 by blocking its dissociation from BiP/GRP78. Finally, they have confirmed their findings in an in vivo model of sepsis. Overall, the findings are interesting and well presented. I have a few minor suggestions for authors to consider:
· How did they establish the doses and duration of exposure of cells to various agents used in this study?
· Are these effects of Sig1R endothelial cells specific or similar role of Sig1R are seen in other cells? At least the authors should comment on it.
· The authors have decided to pretreat cells and mice with different agents which is ok to study the mechanism, but this approach is not therapeutically correct. Did the authors try post sepsis or LPS treatment to see if these agents will have similar effect if the cells or mice treated post LPS/sepsis treatment remains?
· Authors have used one or more asterisks on the figures, but what they mean with these asterisks is not defined.
· Most of the figure legends listed “n=3-4 for each condition” but many figures do not match with these Ns. For example, Fig 11 lists n=4-6 for each condition but the actual dots on bar graphs are more than 6.
